# Evaluation between Biodegradable Magnesium Metal GBR Membrane and Bovine Graft with or without Hyaluronate

**DOI:** 10.3390/membranes13080691

**Published:** 2023-07-25

**Authors:** Marko Blašković, Dorotea Blašković, David Botond Hangyasi, Olga Cvijanović Peloza, Matej Tomas, Marija Čandrlić, Patrick Rider, Berit Mang, Željka Perić Kačarević, Branko Trajkovski

**Affiliations:** 1Department of Oral Surgery, Faculty of Dental Medicine Rijeka, University of Rijeka, Krešimirova 40/42, 51000 Rijeka, Croatia; marko_blaskovic@yahoo.com; 2Dental Clinic Dr. Blašković, Linićeva ulica 16, 51000 Rijeka, Croatia; dmihanovic@sfzg.hr; 3Department of Periodontology, Semmelweis University, 1052 Budapest, Hungary; hangyasidavidbotond@gmail.com; 4Department of Anatomy, Faculty of Medicine, University of Rijeka, Braće Branchetta 20/1, 51000 Rijeka, Croatia; olga.cvijanovic@uniri.hr; 5Department of Dental Medicine, Faculty of Dental Medicine and Health Osijek, Josip Juraj Strossmayer University of Osijek, Crkvena 21, 31000 Osijek, Croatia; matej.tomas@fdmz.hr (M.T.); marija.candrlic@fdmz.hr (M.Č.); 6Botiss Biomaterials, Ullsteinstrasse 108, 12109 Berlin, Germany; patrick.rider@botiss.com (P.R.); berit.mang@botiss.com (B.M.); 7Department of Anatomy, Embriology, Pathology and Pathohistology, Faculty of Dental Medicine and Health Osijek, Josip Juraj Strossmayer University of Osijek, Crkvena 21, 31000 Osijek, Croatia; 8Faculty of Dentistry, Kuwait University, Safat 13110, Kuwait

**Keywords:** immediate implantation, barrier membrane, magnesium membrane, hyaluronate, bovine bone substitute, magnesium screw

## Abstract

Bone substitutes and barrier membranes are widely used in dental regeneration procedures. New materials are constantly being developed to provide the most optimal surgical outcomes. One of these developments is the addition of hyaluronate (HA) to the bovine bone graft, which has beneficial wound healing and handling properties. However, an acidic environment that is potentially produced by the HA is known to increase the degradation of magnesium metal. The aim of this study was to evaluate the potential risk for the addition of HA to the bovine bone graft on the degradation rate and hence the efficacy of a new biodegradable magnesium metal GBR membrane. pH and conductivity measurements were made in vitro for samples placed in phosphate-buffered solutions. These in vitro tests showed that the combination of the bovine graft with HA resulted in an alkaline environment for the concentrations that were used. The combination was also tested in a clinical setting. The use of the magnesium metal membrane in combination with the tested grafting materials achieved successful treatment in these patients and no adverse effects were observed in vivo for regenerative treatments with or without HA. Magnesium based biodegradable GBR membranes can be safely used in combination with bovine graft with or without hyaluronate.

## 1. Introduction

The horizontal and vertical augmentation of severely atrophic alveolar ridges is challenging treatment procedure and requires an adequate bone and soft tissue management [1]. For that reason, various bone grafting materials with different physicochemical properties have been widely used to support bone volume gain [2]. The most commonly used bone substitutes for bone regenerative procedures are bovine derived xenografts [3]. These materials established long-term stability and safety [4,5], especially when treated at very high temperatures, which completely deproteinized the graft and consequently eliminated the risk of any allergic reactions and disease transmission [6]. The high temperature also provides great volume stability by creating a pure hydroxyapatite structure that integrates within the newly produced bone matrix [7]. Furthermore, due to their good mechanical properties and resistance to resorption, xenografts are often used in combination with autogenous bone to achieve better volume stability of the augmented area [8,9]. In addition, it is recommended to mix autogenous bone with bovine xenograft, as this reduces the resorption of autogenous bone by 50% [10]. A recent review on biomaterials used for ridge preservation concluded that xenografts and allografts were the most predictable for providing consistent results for preserving the dimensions of the ridge, but it also indicated that bioactive agents could also have an important role [11].

Nevertheless, there is a constant development of biomaterials to produce the most optimal surgical outcomes by improving osteoblast activity and enhance the bone healing potential [12,13], for example, the biofunctionalization of the graft with HA [14,15]. HA is a large organic polymer naturally present within the human body that provides many essential functions, such as lubrication and is an important part of the wound healing process [16]. The addition of HA has also been shown to significantly increase angiogenesis in vivo [12,17]. Clinically, the addition of HA has been shown to improve implant stability [15] and increase the marginal bone gain and keratinized tissue [14]. 

However, it is necessary to apply guided bone regeneration (GBR) barrier membranes in combination with the grafting materials to prevent the faster proliferating soft tissue cells growing into the defect area [18,19]. The GBR membranes not only serve as a barrier but also to stabilize the formation of the blood clot and the bone grafting material itself [20]. Barrier membranes are divided into non-resorbable (e.g., PTFE, Titanium) and resorbable (e.g., collagen, synthetic polymer) [18,21]. To provide additional support to protect the augmentation, titanium is commonly used to reinforce PTFE membranes [18]. They are highly resistant, which prevents membrane collapse and allows space maintenance for the successful bone regeneration. At the same time, these biomaterials are plastic, which allows good adaptation of the membrane to the defect site. Their main disadvantage is the lack of resorption and, thus, the need for a second surgical procedure to remove the membrane, which can lead to various complications [22,23,24]. 

On the other hand, resorbable membranes have several advantages over non-resorbable membranes, such as avoiding a secondary surgical procedure, which reduces the number of visits and complications [25]. The most commonly used resorbable membranes are made from collagen, which has excellent biocompatibility and tissue integration [26]. However, efforts to extend their degradation time and barrier function, e.g., by chemical crosslinking, can cause foreign body reactions [18,27]. Alternatively, synthetic resorbable options made from polymers can provide a longer stand time but risk fibrous encapsulation and inflammatory reactions. Therefore, a novel biodegradable magnesium metal was developed that combines material strength with resorbability [28,29,30,31,32]. 

Magnesium-based biomaterials are easily resorbed by the human body and have been widely used in various medical fields [28]. The resorption process is based on corrosion and has been successfully applied in producing biodegradable metal membrane as well as fixations screws [28,29,30,31,32]. The magnesium-based GBR membrane has a similar barrier function to the collagen membranes but exerts additional space maintenance properties comparable to titanium-based products [28,29]. It has already been shown that the membrane provides a good bone tissue regeneration and soft tissue healing in combination with an allograft biomaterial [33,34], and a recent review highlighted the many benefits of magnesium metal membranes. The biocompatibility, inflammatory response, and non-toxicity of the magnesium membrane were recently reported [28]. 

The degradation rate of the magnesium membrane can be influenced by surrounding acidity [35,36,37,38,39,40], which can be caused by a remaining infection at the implantation site [41] or degradation by-products of other biomaterials [42]. This could reduce the functional life-span of the membrane in its ability to provide an adequate barrier between the soft and the hard tissues, as well as provide mechanical support to the augmented site. There are currently no publications investigating the interaction of a pure magnesium GBR membrane in combination with bovine graft material with or without hyaluronate.

The aim of this study was to investigate the effect of HA in combination with a bovine bone graft on the efficacy of a recently developed magnesium GBR membrane. Samples were investigated in vitro to evaluate changes in pH and conductivity caused by the HA, which could change the degradation rate of the membrane. In addition, clinical results were retrospectively analysed in two patients that were treated with the magnesium membrane and a bovine bone graft either with or without hyaluronate. 

## 2. Materials and Methods

For the purpose of this study, a biodegradable metal membrane and screws made of magnesium (NOVAMag^®^ membrane and NOVAMag^®^ fixation screws XS, botiss biomaterials GmbH, Berlin, Germany), natural bovine bone grafting material (cerabone^®^, particle size 0.5–1.0 mm, botiss biomaterials GmbH, Germany), and natural bovine bone grafting material with hyaluronate (cerabone^®^ plus, particle size 0.5–1.0 mm, botiss biomaterials GmbH, Germany) were used. 

### 2.1. In Vitro Interactions

Ion release was measured in vitro for the NOVAMag^®^ membrane (NMg-M) size 30 × 40 mm, 1 mL cerabone^®^ (CB) and 1 mL cerabone^®^ plus (CB+), either individually or in combination with each other. Samples were incubated in 40 mL phosphate-buffered saline (PBS) at room temperature. pH and conductivity (EC) measurements were made at 0, 4, 24, 48, 96, 168, and 360 h by pH meter (Seven Excellence, Mettler Toledo, Zagreb, Croatia) and conductivity meter (Seven Excellence, Mettler Toledo, Zagreb, Croatia). The final values were taken as average results obtained from 3 samples per group and a control group of PBS alone was used.

### 2.2. Interactions in Patients

Two patients were retrospectively analysed after being successfully treated with either NMg-M with CB or NMg-M with CB+. The Ethics Committee of the Faculty of Dental Medicine and Health Osijek of the J.J. Strossmayer University Osijek approved the patients’ participation in the study according to the Declaration of Helsinki (Class: 602-01/23-12/05; No. 2158/97-97-10-23-03). Both patients gave their informed consent for participation.

One day before surgery, both patients received an antibiotic regiment of amoxicillin 1000 mg taken twice daily for 7 days. 

Patient 1 was presented with missing mandible bone width in the first left molar region due to previous tooth extraction (Figure 1A). After administrating local anaesthesia (articaine with adrenaline 1:100,000), a full thickness flap was raised and an implant was placed in the edentulous site (NobelReplace^®^ Conical Connection, 4.3 × 8 mm, Kolten, Switzerland) (Figure 1B).

Following the manufacturer’s instructions, 1 mL CB+ was first hydrated with 0.5 mL saline until a sticky ball was obtained and was mixed with autogenous bone (1:1 ratio) to increase the space between the granules. Here, the liquid binding capacity of the hyaluronate enabled easy handling before graft application in Patient 1 (Figure 1C).

A connective tissue graft (CTG) was harvested from the retromolar area for soft tissue correction to increase the width of the keratinized gingiva. In addition, a provisional crown was reshaped to provide more space for the augmented soft tissue (Figure 2).

Patient 2 was presented with missing teeth 35 and 36 (Figure 3A). After administrating local anaesthesia (articaine with adrenaline 1:100,000), a full thickness flap was raised and two implants were placed at the 35, 36 region (NobelReplace^®^ Conical Connection Partially Machined Collar (PMC), 4.3 × 8 mm, Kolten, Switzerland) (Figure 3B). Since two buccal peri-implant dehiscence defects were present after implant placement, it was necessary to perform bone augmentation (Figure 3C). For that reason, 1 mL CB was mixed with autogenous bone (1:1 ratio) to increase the space between the granules and was then applied at this region (Figure 3D). This graft combination did not require pre-hydration procedure of CB as performed with Patient 1. 

Just before graft material application to the recipient site, the NMg-M (15 × 20 mm for Patient 1; 30 × 40 mm for Patient 2) was cut to shape according to the individual needs. In order not to harm the soft tissue, it was also very important to blunt the sharp edges of NMg-M before its use. The membrane was secured buccally by magnesium fixation screws (NOVAMag^®^ fixation screws XS, botiss biomaterials GmbH, Germany) (NMg-XS) size 1.0 mm × 3.5 mm (Figure 1C and Figure 3D). The screws were inserted by pre-drilling at the defect site by using a surgical contra-angel handpiece attached via an adapter (NOVAMag^®^ connector, botiss biomaterials GmbH, Germany). Once the grafting materials were applied and adapted, the NMg-M was bent over and additional screws were inserted lingually in order to achieve graft stabilization (Figure 1D,E and Figure 3E). Due to the reduced width of the keratinized mucosa (KM) and vestibular depth in Patient 2, an apically positioned flap (APP) and a free gingival graft (FGG) were performed from the palatal donor site as shown in Figure 4. Two-layer tension-free sutures were used for closing the flap. Immediately after the surgery, the patients were given 600 mg ibuprofen (Figure 1F and Figure 3F). Patients were instructed to rinse with a 0.12% chlorhexidine solution twice daily for the next two weeks. The sutures were removed two weeks after the surgery and control visits were performed after 4, 8, and 12 weeks. 

## 3. Results

### 3.1. In Vitro Interactions

To evaluate the possible influence of hyaluronate on the degradation of the NMg-M, pH and conductivity measurements were made in vitro for each material individually and in combination with each other over a period of 360 h whilst incubated with PBS. 

Four hours after being incubated in PBS, an increase in pH in all samples was observed, apart from the pure hyaluronate (HA) sample. The HA sample produced and maintained a neutral pH when incubated in PBS. The highest increases at the 4-h time point were for samples that included the NMg-M (Figure 5). A similar pattern was observed after 24 h, where pH values continued to increase. After 50 h, the NMgM-contained samples began to plateau out. 

The CB and CB+ samples had lower pH values in all measurements in comparison to the NMg-M-contained samples, but they were still within an alkaline range. The CB sample constantly provided higher pH values than the CB+ sample. The samples produced a more gradual change in pH, and their pH values were continuing to increase until the end of the test. 

EC measurements showed a different behaviour upon the tested groups (Figure 6). Primarily, the EC in CB+ was much lower than all the other samples and remained constant at all measured timepoints. During the early incubation timepoint, the EC measurements in the CB group were closer to the NMg-M groups; however, a decrease was monitored after 24 h. An individual peak increase in EC for the NMg-M sample occurred at 24 h, after which it reduced back at 48 h and remained relatively consistent. NMg-M combined with either CB or CB+ caused a decrease in EC over the first 24 h. After 24 h, the EC values in NMg-M and CB alone continued dropping, whereas in the NMg-CB+ group, an increase in EC was observed at 96 h before subsequently plateauing out. After 48 h, similar values were reported for the NMg-M and CB+/NMg-M groups. The fluctuations in the conductivity for the samples containing NMg-M are indicative of the formation of degradation products such as an oxide layer on the membrane surface.

### 3.2. Interactions in Patients

Both patients had a successfull hard and soft tissue healing 6 months post surgery, with successful augmentation confirmed via CBCT (Figure 7 and Figure 8). The scan confirmed the bovine graft integration between the newly formed bone matrix and the stable implants osseointegration. 

## 4. Discussion

The aim of this manuscript was to investigate the influence of HA included in a bovine bone graft on the efficiency of a newly developed biodegradable metal GBR membrane. Changes in pH and conductivity of the surrounding aqueous environment can cause an increase in degradation rate for magnesium implants [43]. Therefore, these parameters were monitored for samples incubated in PBS over the period of 360 h at room temperature. Furthermore, the regenerative outcome of two patients treated with either the NMg-M and CB or NMg-M and CB+ were retrospectively analyzed. In both instances, the presence of HA did not have a negative influence on the magnesium membrane. The presence of CB and CB+ did not affect the pH of NMg-M when incubated together, and their conductivity values indicated the formation of degradation products. When implanted together in patients, a successful wound healing and new bone formation was achieved.

Pure bovine-bone-derived substitutes are among the most commonly used biomaterials in the dental regeneration grafting procedures [3]. However, certain differences in their physicochemical properties, mostly due to their processing methods, can have an influence on the final product [2,44]. More specifically, very high temperatures of above 500 °C can be used to deproteinize bovine bone materials and, consequently, eliminate the risk of allergic reactions and disease transmission [6]. Such processing techniques maintain volume stability at the grafted site, but they also support efficient integration with the newly formed bone matrix and a high bone density [7]. 

The naturally occurring polymer hyaluronate is used in many different fields and has been extensively studied for medical applications [45]. Additionally, hyaluronate has been used for many years in dentistry for a variety of indications [46]. Recently, it has also been applied in bovine bone substitutes for biofunctionalization to affect osteoblast activity and enhance bone healing potential [12,13]. Available as a pre-mixed product, the combination of bovine bone substitute with hyaluronate efficiently supported defect grafting and led to a successful sinus augmentation [14,15]. Barrier membranes have been extensively studied in order to prevent soft tissue infiltration within the newly formed bone during the GBR process [18]. They can be classified as non-resorbable (e.g., PTFE, Titanium) and resorbable (e.g., collagen, synthetic polymer) membranes. Each type of membrane has its own advantages and disadvantages, for instance, PTFE membranes can be successfully used in open healing [47,48,49,50]. Differences in mechanical and physicochemical properties of the PTFE membranes may potentially affect the clinical outcomes of dental GBR procedures [51]. Titanium can be used to reinforce PTFE membranes and is also produced as various meshes or cages that provide space maintenance at the grafting site [18]. However, in all cases, titanium membranes must be removed from the surgical site, which may require a second surgery and cause various complications such as exposure or infection [24]. 

For these reasons, resorbable membranes are preferred nowadays. They exert several advantages over non-resorbable membranes, among which are easier handling, not requiring membrane removal, lower patient morbidity rate, and improved soft tissue healing [25]. The most commonly used resorbable membranes are made from collagen; however, their variations in degradation time can affect the barrier function, which is why some are processed by chemical crosslinking in order to extend the degradation, which can cause foreign body reactions [18,27]. Alternatively, polymeric membranes, such as those composed of PDLLA, have a long term stability within the body; however, fibrous encapsulation and delayed foreign body reactions have been reported [21,52]. Therefore, the lack of an ideal GBR membrane led to the creation of a novel biodegradable metal variety [28,29,30,31,32]. 

Magnesium metal is completely resorbed by the human body without any toxic residual. Its degradation is based on corrosion process where the magnesium is oxidized, magnesium ions are released, hydrogen atoms from the water are reduced, and as a result, magnesium hydroxide layer and hydrogen gas are formed [28,29,30,31,32]. This magnesium hydroxide layer is then attacked by the sodium chloride, which forms soluble magnesium chloride salt that further contributes the corrosion and thus continuous resorption [28].

Magnesium is already being used for cardiovascular stents, tracheal stents, orthopedic screws, osteosynthesis systems (cranio-maxillofacial surgery), and bone repair materials, such as fracture plates [28]. More recently, magnesium metal has been used for biodegradable metal membranes and fixations screws in dentistry [28,29,30,31,32]. The new magnesium membranes not only provided comparable barrier function to the collagen membranes but also supported space maintenance and the formation of new bone [28,29]. The magnesium fixation screws were also reported to undergo a similar degradation and were replaced by new bone [30,31]. 

The combination of different bone regenerative products and their influence on their respective regenerative properties needs to be considered, especially with regard to resorbable materials. Previous in vitro studies indicated that the combination of HA with magnesium implants does not pose a problem [53,54], which was further proven by an in vivo that combined a HA hydrogel inside cannulated screws implanted into the femur of rats [55]. In this instance, the HA hydrogel was used to reduce the pH of the local environment caused by the degradation of the magnesium fixation screw.

However, no recent study investigated the potential interaction between the biodegradable magnesium metal GBR membrane and bovine graft with or without hyaluronate. As expected, in vitro results showed an increase in pH in the CB group due to the release of OH ions [56]. According to the results obtained, the pH values of the CB + group were slightly lower than those of the CB group, caused by the presence of HA that produced a neutral pH when incubated in PBS (Figure 5). The released OH ions interacted with the hyaluronate, which, in turn, caused some neutralization effect. Additionally, the volume of HA used in the test corresponded to the volume of HA used in the commercial product cerabone^®^ plus. Therefore, it is likely that, at higher concentrations, HA will have a greater influence on the surrounding environmental pH. 

Moreover, the presence of CB and CB + had no effect on the NMg-M, most likely due to the higher release of OH ions and the higher pH values caused by the NMg-M itself within the first 24 h. The pH measurement also provides an indication as to the degradation of the NMg-M magnesium metal and the release of hydrogen atoms, which occurs as part of the degradation process in an aqueous environment [39]. Therefore, the hyaluronic acid did not induce an acidic environment that could increase the rate of magnesium degradation.

On the other hand, the release of calcium and magnesium ions by the samples induced different conductivity profiles (Figure 6). It seems that the presence of hyaluronate in the CB+ group reacts with the released calcium and the EC profile was much closer to the control rather than to the CB group. On contrary, in the CB group, the EC was observed to be higher until 24 h with a sudden drop at 48 h. The NMg-M profile showed similar initial EC values with CB, followed by a significant increase at 24 h, indicating much faster material degradation. However, this EC increase was also followed by a drop after 48 h. This is most probably due to the initial burst of calcium and magnesium release that later lose their elemental state [57]. On the contrary, the combination samples had different behavior as EC started increasing after 24 h and remaining stable after 96 h. Precisely, there was a decrease in the first 24 h, then an increase, that nearly reached the starting points for both combination samples. This would suggest a synergistically delayed calcium and magnesium release without any further mutual interaction, except the slight EC drop and delay caused by the hyaluronate. 

Both patients had successful healing and, 6 months later, a control CBCT was performed (Figure 7 and Figure 8). The images showed homogeneous osseointegration of the bone graft within the newly formed bone, which enabled implants stability. Similar behavior has already been reported where the magnesium membrane was used for rebuilding both buccal/palatal walls via a “shield” to provide mechanical support [33,58]. Also, it would be challenging to verify if the magnesium chloride salt layer, which is an NMg-M degradation product, imitates a calcium phosphate mineral during the remodeling phase and serves as a 2D scaffold that turns into a cortical layer. 

Future studies on bovine graft with or without HA with magnesium metal GBR membrane could focus on improving their properties to accelerate bone healing and increase the durability of dental implants. Mixing HA with bovine graft could improve graft stability and enhance integration of the xenograft into the tissue. This area of research could result in innovative approaches for bone tissue regeneration. 

## 5. Conclusions

Within the limitations of this study, no adverse interaction between biodegradable magnesium metal GBR membrane and bovine graft with or without hyaluronate was monitored/reported. The implantation of a magnesium metal membrane in combination with bovine bone substitute with or without hyaluronate was observed, resulting in a successful treatment in patients. Furthermore, no significant change in pH or conductivity was observed in vitro that would indicate a negative effect on healing. Nevertheless, further *clinical* studies and a larger number of patients are needed to verify our observations. 

## Figures and Tables

**Figure 1 membranes-13-00691-f001:**
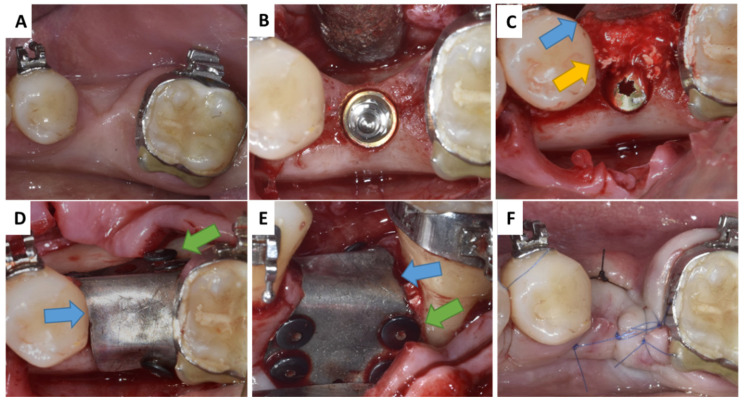
Surgery protocol - Patient 1: (**A**,**B**) Missing mandibular bone; (**C**) application of grafting material; (**D**,**E**) coverage with magnesium metal GBR membrane; (**F**) tension-free suturing. Blue arrow—magnesium metal membrane; green arrow—magnesium metal fixation screw; yellow arrow—natural bovine bone grafting material with hyaluronate/autogenous bone mix.

**Figure 2 membranes-13-00691-f002:**
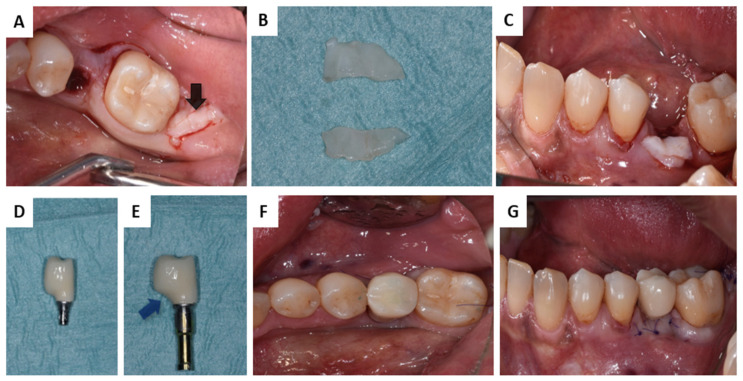
CTG of Patient 1: (**A**) Harvesting donor site (black arrow); (**B**) CTG extraoral de-epitelisation; (**C**) adjusting the CTG; (**D**) provisional crown; (**E**) reshaped provisional crown (blue arrow); (**F**) screw retained provisional crown—occlusal view; (**G**) screw retained provisional crown—buccal view.

**Figure 3 membranes-13-00691-f003:**
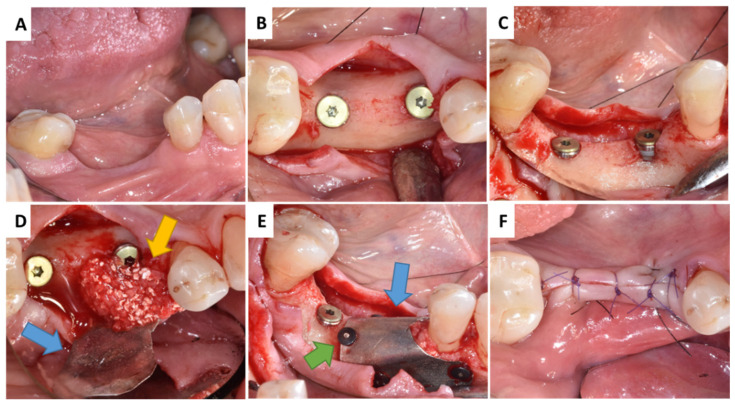
Surgery protocol of Patient 2: (**A**) Missing teeth 35 and 36; (**B**,**C**) implants placement; (**D**) application of grafting material; (**E**) coverage with magnesium metal GBR membrane; (**F**) tension-free suturing. Blue arrow—magnesium metal membrane; green arrow—magnesium metal fixation screws; yellow arrow natural bovine bone grafting material with hyaluronate/autogenous bone mix.

**Figure 4 membranes-13-00691-f004:**
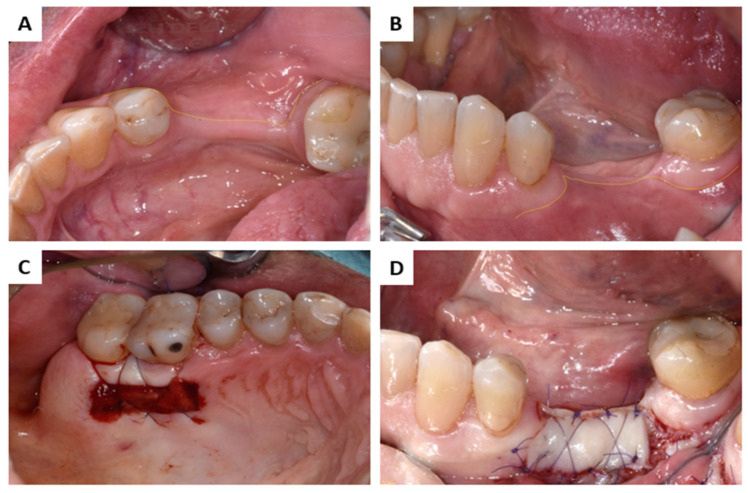
AFP + FGG of Patient 2: (**A**) Reduced KM width; (**B**) reduced KM and vestibular depth; (**C**) palatal donor site; (**D**) APF + FGG.

**Figure 5 membranes-13-00691-f005:**
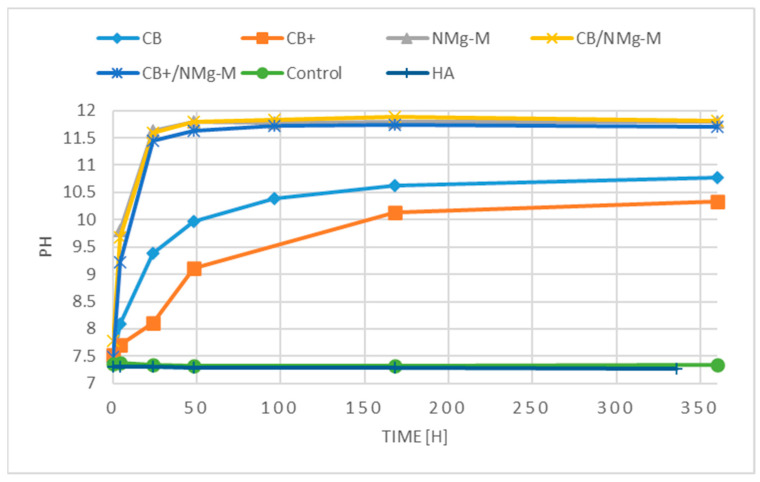
Change in pH values of PBS where samples were incubated. X-axis is hours, y-axis is pH values.

**Figure 6 membranes-13-00691-f006:**
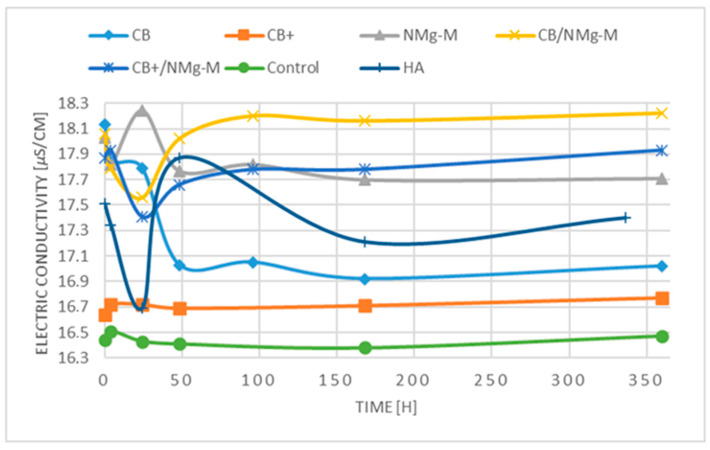
Change in electric conductivity of the PBS where samples were incubated. X-axis is hours, y-axis is µS/cm values.

**Figure 7 membranes-13-00691-f007:**
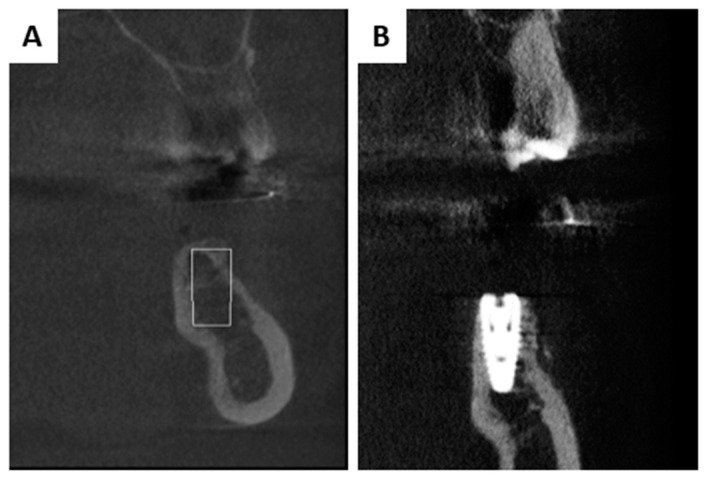
CBCT before (**A**) and after six months of augmentation (**B**) for Patient 1. CBCT shows homogeneous osseointegration of the bone graft within the newly formed bone.

**Figure 8 membranes-13-00691-f008:**
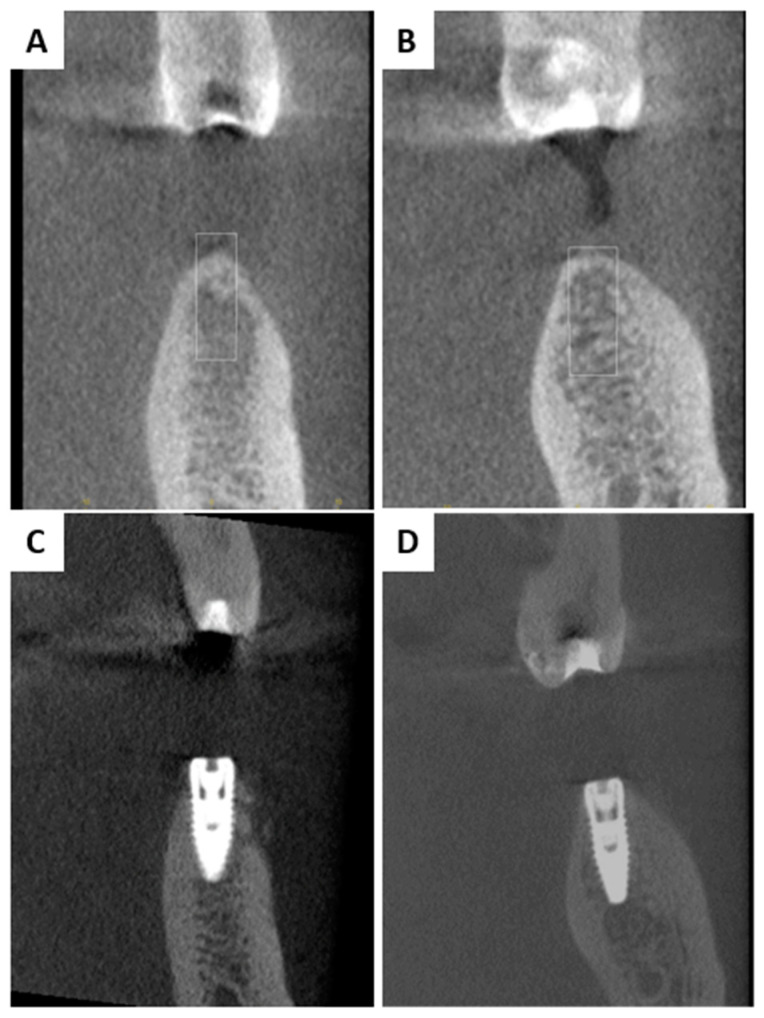
CBCT before (**A**,**B**) and after six months of augmentation (**C**,**D**) for Patient 2. CBCT shows homogeneous osseointegration of the bone graft within the newly formed bone.

## Data Availability

The data presented in this article are available on request from the corresponding authors.

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
