# Peer review of "Evaluation between Biodegradable Magnesium Metal GBR Membrane and Bovine Graft with or without Hyaluronate"

_membranes, 2023, doi:10.3390/membranes13080691_

Round 1
Reviewer 1 Report
The authors perform an analysis regarding pH and conductivity of magnesium-based membranes and screws and natural bovine graft material with or without hyaluronate. Furthermore, the outcomes from one patient treated with magnesium membranes and bovine graft material without hyaluronate and another patient treated with magnesium membranes and bovine graft material with hyaluronate were retrospectively analyzed. The manuscript is well organized; the in vivo part of the study is well described, however, some points must be improved. The introduction section could be improved by adding recently published papers on the topic PMID: 34826029
- PMID: 34825280
Biodegradable membranes with similar characteristics are PDLLA membranes (PMID: 29236060); the authors could make a comparison between the two materials in the discussion section.
The main concern of this paper is about the in vitro part: the authors should clearly state the rationale to measure only pH and conductivity of the materials used in order to evaluate the interactions between them.
In my opinion, the conclusions, both in the abstract and in the main text, should be rephrased in order to make them more responsive to the obtained results.
The manuscript should be rearranged in order to avoid too general conclusions, that cannot be supported by the obtained results.
Reviewer 2 Report
he use of magnesium-based membranes is quite recent, andfrom what I could assess there are 5 publications, 4 of which are
only "in vitro" and another that has an "in vitro" and "in vivo" part. Therefore, evaluating the behavior of a membrane based on an "in vitro" study
and the report of two cases does not seem to me to be something that
scientifically can support or even indicate the use of a material or membrane. Apparently the authors wanted to report 2 cases with the use of a magnesium-based
membrane and the effect of its degradation or biocompatibility through 2 cases.
Unfortunately from my point of view this is poor and insufficient.
The use of animal models is of paramount importance so that we can validate
products in terms of: inflammatory reaction, foreign body type reaction,
degradability time, mechanical resistance, and in this case,
even evaluating distant organs due to possible contamination with the metal. I believe that this work is not in the profile of the magazine
Reviewer 3 Report
This study “Evaluation between biodegradable magnesium metal GBR membrane and bovine graft with or without hyaluronate” is interesting. A few questions should be considered:
1. Compared with the current clinical methods, what’s the advantage of the use of bovine graft? What’s the advantage of use of hyaluronate compared with “without hyaluronate”? Does this study show the advantages of Mg-based biodegradable GBR membranes in combination with bovine graft with or without hyaluronate?
2. The conclusions suggest that the add of hyaluronate did not change the pH or conductivity in vitro, is it possible that the reason is the change of pH value or conductivity in vitro is related with the amount of added hyaluronate ? Does the current added amount show obvious advantages compared with “without hyaluronate”?
4. In the future study, will the authors intend to choose some challenging clinical cases?
1. Please polish the whole text by native English speaker.
Round 2
Reviewer 1 Report
The authors improved the manuscript following my suggestions.
I think the paper can be considered for publication.
Author Response
it is attached

Reviewer 2 Report
Dear Editor The changes in the text by the authors were clear. However, with regard to theobjective of the work, it remains the same, that is, discuss the
function of the limbs with the report of two cases.
The authors reorganized the text in order to show "in vitro" results. However, there are no
preclinical results that establish that this
material will allow a good biological response, such as: not being toxic,
not having exacerbated inflammatory responses, among other criteria.
If this publication were only about case reports ok, however for a research journal
I believe it does not fit
Author Response
it is attached
